# Spatio-temporal Decoupled Knowledge Compensator for Few-Shot Action Recognition

## Abstract

Few-Shot Action Recognition (FSAR) is a challenging task that requires recognizing novel action categories with a few labeled videos. Recent works typically apply semantically coarse category names as auxiliary contexts to guide the learning of discriminative visual features. However, such context provided by the action names is too limited to provide sufficient background knowledge for capturing novel spatial and temporal concepts in actions. In this paper, we propose **DIST**, an innovative **D**ecomposition-**i**ncorporation framework for FSAR that makes use of decoupled **S**patial and **T**emporal knowledge provided by large language models to learn expressive multi-granularity prototypes. In the decomposition stage, we decouple vanilla action names into diverse spatio-temporal attribute descriptions (*i.e.*, action-related knowledge). Such commonsense knowledge complements semantic contexts from spatial and temporal perspectives. In the incorporation stage, we propose Spatial/Temporal Knowledge Compensators (SKC/PKC) to discover discriminative object- and frame-level prototypes, respectively. In SKC, object-level prototypes adaptively aggregate important patch tokens under the guidance of spatial knowledge. Moreover, in TKC, frame-level prototypes utilize temporal attributes to assist in inter-frame temporal relation modeling, further understanding diverse temporal patterns in videos. These learned prototypes at varying levels of granularity thus provide transparency in capturing fine-grained spatial details and dynamic temporal information, so as to enable accurate recognition of both appearance-centric and motion-centric actions. Experimental results show DIST achieves state-of-the-art results on four standard FSAR datasets (*i.e.*, Kinetics, UCF101, HMDB51 and SSv2-small). Full code will be released.

## 1 Introduction

With deep learning advancements [1, 2, 3, 4], significant progress has been made in the field of action recognition [5, 6, 7, 8, 9] recently. However, this success relies heavily on a large amount of manually-labeled samples, which are time-consuming and expensive to acquire. To alleviate the data-hunger issue, considerable works [10, 11, 12, 13, 14] have turned their attention to few-shot action recognition (FSAR), where there exist base action classes (seen) with a large volume of training examples, and novel action classes (unseen) with unlabeled samples. FSAR learns feature representation from base action classes, and then evaluates its generalization ability on novel action classes.

Modern FSAR solutions [15, 16, 17, 18, 19, 20, 21] are largely built upon metric-based meta-learning paradigm [22], where the model learns class (prototype) representation and performs prototype-query matching with respect to predefined or learned distance metrics. Among them, the top-leading methods [11, 15, 23, 13] directly extract class-related spatio-temporal feature representation from raw visual signals. Though impressive, these methods lack a basic grasp of explicit action knowledge, struggling to learn new concepts in action classes, particularly under data-limited conditions. Recent works [24, 21, 25] transfer knowledge from pre-trained vision-language models [26, 27] (*e.g.*, CLIP [27]) to enhance FSAR model capability. However, these methods typically apply semantically coarse or ambiguous category names as auxiliary context information to compensate for visual features. Such context provided by the action names is too limited to provide enough background knowledge for video action understanding [28, 29, 30].

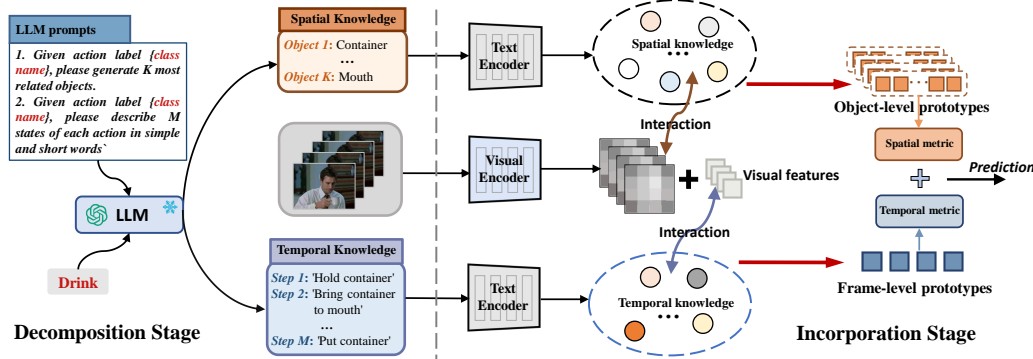

Figure 1: **Our main idea**. Our approach decomposes category names into diverse spatio-temporal knowledge, and makes use of the decoupled knowledge to learn object-/frame- level prototypes, respectively.

To address this issue, we study how to effectively collect and leverage action-related commonsense knowledge provided by large language models (LLMs), compensating for visual information and thus enhancing the few-shot learning capacity. The LLM serves as a knowledge base to provide action-related background commonsense descriptions from complementary spatial and temporal perspectives, which we refer to as decoupled "*spatio-temporal prior knowledge*". Compared to vanilla categories, spatio-temporal prior knowledge **i)** makes up for missing context information to achieve semantic completeness, and **ii)** transforms unseen categories into known commonsense descriptions, easily interpreted by pre-trained language models.

In light of the above, we develop a novel decomposition-incorporation framework for FSAR: **DIST**, which firstly decomposes vanilla category names into diverse spatio-temporal attribute descriptions, and then incorporates decoupled commonsense knowledge and visual features to guide the learning of object-level and frame-level prototypes. As shown in Fig. 1, for the **decomposition stage**, we make use of LLMs, to generate action-related commonsense descriptions (*i.e.*, external contextual information and different steps of action) from coarse category names. Such comprehensive descriptions complement semantic contexts from spatial and temporal perspectives. In the **incorporation stage**, we propose Spatial/Temporal Knowledge Compensators (SKC/TKC), which incorporate decoupled prior knowledge and visual features to form discriminative object-level (spatial) and frame-level (temporal) prototypes, respectively. Specifically, SKC aggregates important patches into compact object prototypes by patch-level cross-attention within each frame, and further guides object-level prototype learning with the assistance of spatial prior knowledge. These object-level prototypes filter out video noise and focus on informative image patches, which refer to the most class-related ones and correspond to key entities. Meanwhile, TKC captures the temporal relationships between frame-level prototypes through inter-frame interaction, and then enables these prototypes to aggregate essential semantic information from temporal prior knowledge. The learned different-level prototypes can capture fine-grained spatial details and dynamic temporal information, respectively, so as to yield more accurate action recognition results.

Overall, our contributions are summarized as follows: ❶ We pioneer the early exploration in making use of action-related prior knowledge for FSAR, and to achieve this end, we construct background commonsense descriptions provided by LLMs in a spatiotemporal-decoupled manner. ❷ We propose a novel decomposition-incorporation framework that decouples category names into diverse spatial and temporal prior knowledge, and then incorporates them and visual features to learn object-level and frame-level prototypes in a dual way. ❸ We design Spatial/Temporal Knowledge Compensators (SKC/TKC) that inject decoupled prior knowledge into different-level prototypes to capture fine-grained spatial details and dynamic temporal information.

To the best of our knowledge, we make the pioneering effort to explore the application of diverse spatio-temporal prior knowledge in FSAR, aiming to effectively provide semantic contexts for the learning of different-level prototypes from multiple perspectives. Different from simply combining rich LLM-generated descriptions (*i.e.*, only one sentence about actions) and frame-level features in a single branch, our framework conducts customized feature interaction for different branches to incorporate i) patch-level features and spatial knowledge; and ii) frame-level features and temporal knowledge, respectively (see Table 5c in §4.4). Such a framework can discover fine-grained spatial patterns and dynamic temporal patterns, hence producing more accurate few-shot results. To

comprehensively evaluate our method, we conduct experiments on four gold-standard datasets (*i.e.*, HMDB51 [31], UCF101 [32], kinetics100 [33], and SSv2-small [34]). We empirically prove that DIST surpasses all existing state-of-the-arts and yields solid performance gains (**1.7%**-**6.8%** accuracy) under the 5-way 1-shot setting. Furthermore, we perform thorough ablation studies to dissect each component, both quantitatively and qualitatively. Our full implementation will be released.

## 2 RELATED WORK

**Few-shot Image Classification.** The objective of few-shot image classification [35, 36] is to recognize new categories with a small number of annotated samples. Existing methods can be roughly categorized into three groups: **i)** *Augmentation-based* methods [2, 37, 38] exploit various augmentation strategies to alleviate the data scarcity dilemma, mainly including spatial deformation [39] and feature augmentation [40, 41]; **ii)** *Optimization-based* methods [42, 43, 44, 45] learn optimization states, like model initialization [42, 43] or step sizes [44, 45], to update models with a few gradient steps; and **iii)** *Metric-based* methods [46, 47, 22, 48, 49, 50] learn a class representation (prototype) by averaging embeddings belonging to the same class, and predict query (*i.e.*, test sample) labels with respect to predefined [46, 47, 22, 48] or learned [49, 50] distance metric. Our work is more closely related to the metric-based methods [47, 22], whereas we focus on few-shot action recognition – a more challenging task that requires handling videos encompassing a wealth of temporal information due to common and distinct patterns in nearby frames [51].

**Few-shot Action Recognition (FSAR).** FSAR is a challenging task with the goal of recognizing previously unseen action classes (*i.e.*, query class) with a few labeled videos. Existing FSAR methods [52, 53, 54, 13, 15] mainly belong to the metric-based meta-learning paradigm [47], which learns class (prototype) representation and performs prototype-query matching based on the learned distance metrics. These methods are mainly devoted to feature representation learning [15, 13] and matching strategy exploration[52, 55, 56, 11, 18, 16, 13]. As a primary step, *feature representation learning* helps models to learn expressive spatio-temporal features for further matching process. Recent appoarches [15, 13] model temporal features through temporal attention operations [15] or more detailed temporal-patch and temporal-channel interaction [13], and further exploit low-level spatial features by patch-level information interaction within each frame or across frames [57]. Some others make use of video features in a whole task (*i.e.*, episode) to extract relevant discriminative patterns [13, 16, 58] by a graph neural network [13] or attention relation modeling [16]. For *matching strategy exploration*, early works [52, 55, 56] aggregate the frame features into a single video representation for video-level feature matching. Though straightforward, these methods suffer from suboptimal performance due to neglecting the temporal cues in videos. To address this limitation, the following approaches [11, 18, 16, 13, 23] devise various temporal alignment metrics for frame-level feature matching, *e.g.*, frame-level alignment [11, 16], segment-level alignment [18], and even frame-to-segment alignment [23] that is also common in realistic video matching.

Recent works [24, 59, 21] transfer knowledge from pre-trained vision-language models (*e.g.*, CLIP [27])toenhance FSAR model capability. Though promising, they heavily rely on semantically coarse or ambiguous category names as semantic source to provide action-related context. Such context is insufficient to offer enough background knowledge for video understanding. In contrast, our DIST represents the first effort in FSAR to decouple class names into diverse spatio-temporal attribute descriptions (*i.e.*, action-relevant knowledge) to complement semantic contexts. More significantly, such acquired decoupled knowledge is further injected into visual features to learn object- and frame-level prototypes in a dual way, so as to enable more accurate action recognition.

**Few-shot Learning with Semantic Information.** Recent works on few-shot learning [60, 61, 62] integrate semantic information (provided by class labels) and visual information (extracted from visual observations) to represent a novel class. Based on the levels at which modality information fusion occurs, these methods can be roughly categorized into three groups: **i)** *Prototype-level* methods [63, 64] model class (prototype) representation as a combination of visual prototypes and semantic prototypes obtained through word embeddings of class labels by attention mechanism [64] or adaptive fusion mechanism [63]; **ii)** *Classifier-level* methods [60] enable classifiers to predict novel categories by incorporating auxiliary semantic information acquired from a graph convolutional network [65]; and **iii)** *Extractor-level* methods [66, 62] consider semantic information as prompts to tune the feature extractor, allowing the feature extractor to better focus on class-specific features.

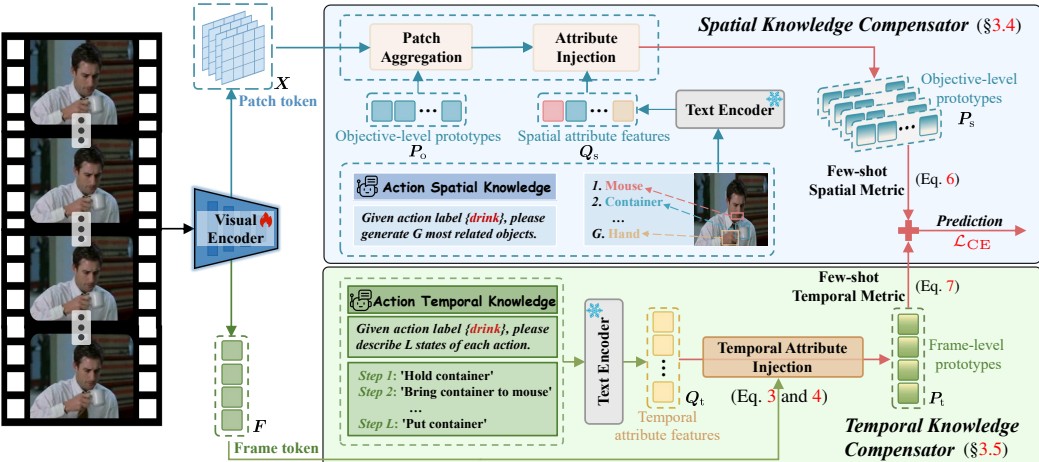

Figure 2: **Overview of DIST**. Video inputs are first processed by the visual encoder of CLIP to obtain initial patch-level and frame-level features. Then, we leverage LLM to decompose vanilla action names into action-related background knowledge. Furthermore, SKC/TKC incorporate decoupled prior knowledge and visual features to form discriminative object- and frame-level prototypes for spatial/temporal matching. Finally, we can combine spatial and temporal matching results to obtain the merged query prediction.

Though impressive, they [63, 66, 62] typically directly apply semantically coarse category names as auxiliary information at different levels to compensate for visual features, lacking high-quality background knowledge to discover novel visual concepts, therefore struggling with adapting to unseen categories. Our contribution is orthogonal to previous studies, as we advance FSAR regime in the aspect of collecting and leveraging high-quality spatio-temporal prior knowledge. The LLM serves as a knowledge base to provide action-related background commonsense descriptions (*i.e.*, contextual information and different steps of actions). Then our work makes smart use of prior knowledge to reduce redundant visual features and enhance the semantic distinction of different class prototypes.

## 3 METHOD

### 3.1 PROBLEM FORMULATION

The goal of FSAR is to classify unlabeled test videos given a few (*e.g.*, one or five) samples per class. Under the few-shot setting, the model learns a feature representation on training classes $\mathcal{D}_{base}$ and is evaluated on testing classes $\mathcal{D}_{novel}$ to emphasize its generalization ability on novel categories, where $\mathcal{D}_{base} \cap \mathcal{D}_{novel} = \emptyset$. In the training stage, follwing [67, 16, 13], we train a few-shot learning model in an episodic way. Here, each episode (*i.e.*, a standard $M$-way $K$-shot episode task) is formed by sampling $M$ categories from $\mathcal{D}_{base}$. The $M$-way $K$-shot task consists of $K$ labeled videos per class as the support set $\mathcal{S}$, and a fraction of the rest samples from $M$ classes as the query set $\mathcal{Q}$. The episodic training for FSAR is achieved by minimizing, for each episode, the loss of the prediction on samples in the query set, given the support set. In the inference stage, we randomly sample episode tasks from $\mathcal{D}_{novel}$ for evaluation, and report average results over multiple episode tasks.

### 3.2 OVERALL FRAMEWORK

We introduce DIST, which collects and leverages action-related commonsense knowledge provided by LLMs to guide the learning of object- and frame-level prototypes for FSAR (Fig. 2). DIST consists of visual/text encoders and two knowledge compensators. The model takes the RGB frame sequence of length $T$ and corresponding action names as input. We first utilize the visual encoder of CLIP [27] to get frame-level (*i.e.*, class token in each frame) and patch-level feature representation, *i.e.*, $\boldsymbol{F} \in \mathbb{R}^{T \times C}$ and $\boldsymbol{X} \in \mathbb{R}^{T \times P \times C}$, where $P$ is the number of tokens in each frame. Besides, we prompt LLM with corresponding action names to generate decoupled spatial and temporal commonsense descriptions (§3.3). These descriptions are fed into frozen text encoder of CLIP to obtain spatial and temporal attribute features, *i.e.*, $\boldsymbol{Q}_s \in \mathbb{R}^{G \times C}$ and $\boldsymbol{Q}_t \in \mathbb{R}^{L \times C}$. We further design two complementary modules to make use of decoupled spatio-temporal attributes: 1) Spatial Knowledge Compensator (SKC) (§3.4) injects spatial attributes into patch-level features to explicitly learn com-

pact object-level prototypes for object-level prototype matching; 2) Temporal Knowledge Compensator (TKC) (§3.5) incorporates temporal attributes and frame-level features to inform frame-level prototypes for frame-level prototype matching. Finally, we can combine spatial and temporal feature matching scores to obtain the merged query prediction.

### 3.3 DECOUPLED SPATIO-TEMPORAL ATTRIBUTE GENERATION

**Spatial Attribute Generation.** Naive category names provided limited commonsense knowledge to focus on action-related spatial contexts. Thus we make use of prior knowledge in LLMs [68, 69] to generate detailed and informative spatial attributes for each category, *i.e.*, action-related object instances and environment. Specifically, taking the action category "*drink*" as an example, to obtain spatial attribute descriptions, we prompt ChatGPT [69] by "*Given action label {drink}, please generate {G} most related objects for each class.*", where $G$ is empirically set to 6. This prompt returns a set $\mathcal{A}$ with $G$ spatial attribute descriptions, such as "*container; mouth; hand; ...*". Then we encode these spatial attributes via frozen CLIP text encoder to get spatial attribute features $\boldsymbol{Q}_{\mathrm{s}} \in \mathbb{R}^{G \times C}$.

**Temporal Attribute Generation**. Prior researches [24, 59] apply semantically coarse category names as auxiliary information to guide temporal feature learning. However, such context provided by action names is too limited to provide enough temporal context for action recognition. Thus, we propose to utilize the abundant prior knowledge in LLMs [68, 69] to expand the coarse action names. Temporal attribute descriptions generated by LLM are a collection of multiple atomic actions, which describe the temporal evolution of an action. Concretely, taking the action category "*drink*" as an example, to obtain temporal attribute descriptions, we prompt ChatGPT [69] by "*Given action label {drink}, please describe {L} states of each action in simple and short words.*", where we empirically set $L$ to 3. This prompt always returns a set $\mathcal{B}$ with $L$ temporal attribute descriptions, such as "*Hold container; Bring container to mouth; Put container; ...*", which decompose one action class into multiple atomic actions in a step-by-step manner. Then we adopt the off-the-shelf text encoder of CLIP to encode these descriptions and obtain temporal attribute features $\boldsymbol{Q}_{\mathrm{t}} \in \mathbb{R}^{L \times C}$.

### 3.4 SPATIAL KNOWLEDGE COMPENSATOR

Previous methods exploit spatial features by patch-level information interaction within each frame or across frames [15, 57]. However, this leads to two issues: 1) Too many irrelevant patch tokens bring redundant information, interfering with further spatial feature matching; 2) They fail to focus on important objects without the guidance of spatial prior knowledge. Therefore, we investigate how to better incorporate spatial attributes (§3.3) and patch-level visual features into compact object-level prototypes to highlight potential target objects. To this end, as showcased in Fig. 3, proposed Spatial Knowledge Compensator (SKC) summarizes discriminative spatial patterns via aggregating patch-level features into compact object-level prototypes (*i.e.*, patch aggregation), and further delivers the union of spatial attribute knowledge and such object-level prototypes to enhance learned spatial patterns (*i.e.*, attribute injection).

**Patch Aggregation.** We first introduce a set of learnable object-level prototypes to aggregate image content and highlight potential target objects. The prototypes are randomly initialized embeddings and represented as $\boldsymbol{P}_{\mathrm{o}} \in \mathbb{R}^{N \times C}$, where $N$ is the number of object prototypes. Firstly, a self-attention layer is adopted for the $N$ object prototypes to interact with each other in each frame. Then, these prototypes aim to adaptively aggregate action-related or object-related key patches in a sparse manner by patch-level cross-attention within each frame. Specifically, for patch tokens $\boldsymbol{X}^{l} \in \mathbb{R}^{P \times C}$ in $l$-th frame, the process can be defined as:

$$\hat{\boldsymbol{P}}_{\mathrm{o}} = \mathrm{Softmax}(\boldsymbol{P}_{\mathrm{o}} \boldsymbol{K}_{\mathrm{p}}^{\top}) \boldsymbol{V}_{\mathrm{p}} + \boldsymbol{P}_{\mathrm{o}}, \tag{1}$$

where $\boldsymbol{K}_{\mathrm{p}}$ and $\boldsymbol{V}_{\mathrm{p}}$ are the linear transformation features of patch tokens $\boldsymbol{X}^{l}$. This allows the object-level prototypes to capture discriminative spatial patterns.

**Attribute Injection.** To further encourage object prototypes to focus on action-related spatial context information, we deliver the union of spatial attribute knowledge and such object-level prototypes to discover fine-grained spatial patterns via the attention mechanism as follows:

$$\boldsymbol{P}_{\mathrm{s}} = \mathrm{Softmax}(\hat{\boldsymbol{P}}_{\mathrm{o}} \boldsymbol{K}_{\mathrm{q}}^{\top}) \boldsymbol{V}_{\mathrm{q}} + \hat{\boldsymbol{P}}_{\mathrm{o}}, \tag{2}$$

where $\boldsymbol{K}_{\mathrm{q}}$ and $\boldsymbol{V}_{\mathrm{q}}$ are the linear transformation features of spatial attribute features $\boldsymbol{Q}_{\mathrm{s}}$, $\boldsymbol{P}_{\mathrm{s}} \in \mathbb{R}^{N \times C}$ is learned diverse object prototypes. Note that the operation of object prototypes in each frame is

Figure 3: Illustration of Spatial Knowledge Compensator (SKC). SKC aims to learn discriminative object-level prototypes in a sparse aggregation manner via patch aggregation and attribute injection.

the same. By exchanging information for visual features and attribute features respectively, learned object-level prototypes filter out redundant information in videos and capture spatial details.

## 3.5 TEMPORAL KNOWLEDGE COMPENSATOR

How to incorporate temporal attribute features and frame-level features is essential for better FSAR performance since temporal prior knowledge can enable the model to understand dynamic semantics. Thus, our Temporal knowledge Compensator (TKC) aggregates essential semantic information by injecting temporal prior knowledge into visual features.

Specifically, we obtain global semantic vector $\boldsymbol{p}_g \in \mathbb{R}^{1 \times C}$ by pooling temporal attribute features, and add it to frame-level features $[\boldsymbol{f}_1, \boldsymbol{f}_2, ..., \boldsymbol{f}_T] \in \mathbb{R}^{T \times C}$:

$$\boldsymbol{F}_q = [\boldsymbol{f}_1 + \boldsymbol{p}_g, ..., \boldsymbol{f}_T + \boldsymbol{p}_g], \tag{3}$$

where $\boldsymbol{F}_q \in \mathbb{R}^{T \times C}$ is the obtained frame-level prototypes, which incorporate overall semantic information. The frame-level prototypes further aggregate temporal context information from temporal prior knowledge via vision and attribute cross-attention mechanism. Then the frame prototypes are fed into the temporal transformer [16] to capture the temporal relationships between frame-level prototypes. This is given by

$$\boldsymbol{P}_t = \texttt{Tformer}(\text{Softmax}(\boldsymbol{F}_q \boldsymbol{K}_t^\top) \boldsymbol{V}_t + \boldsymbol{F}_q), \tag{4}$$

where $\boldsymbol{K}_t$ and $\boldsymbol{V}_t$ are the linear transformation features of temporal attribute features $\boldsymbol{Q}_t$, $\texttt{Tformer}$ is the temporal transformer [16], $\boldsymbol{P}_t \in \mathbb{R}^{T \times C}$ is the frame-level prototypes capturing action dynamic information. In this way, the learned frame-level prototypes can adaptively perceive temporal changes and encode the action temporal context with the guidance of temporal knowledge.

## 3.6 FEW-SHOT METRIC

**Few-shot Spatial Metric.** To conduct spatial feature matching between videos, we propose an object-level prototype matching strategy based on the bidirectional Hausdorff Distance [16], which calculates the distances between query object-level prototypes and support object-level prototypes from the set matching perspective. Specifically, given query object-level prototypes $\boldsymbol{P}_s \in \mathbb{R}^{T \times N \times C}$ and support object-level prototypes $\hat{\boldsymbol{P}}_s \in \mathbb{R}^{T \times N \times C}$, we apply a bidirectional Mean Hausdorff metric to obtain a frame-level distance matrix $\hat{\boldsymbol{D}} = [d_{ij}]_{T \times T} \in \mathbb{R}^{T \times T}$ as:

$$d_{ij} = \frac{1}{N} \sum_{\boldsymbol{p}_{i,k}^s \in \boldsymbol{p}_i^s} (\min_{\hat{\boldsymbol{p}}_{j,l}^s \in \hat{\boldsymbol{p}}_j^s} \|\boldsymbol{p}_{i,k}^s - \hat{\boldsymbol{p}}_{j,l}^s\|) + \frac{1}{N} \sum_{\hat{\boldsymbol{p}}_{j,l}^s \in \hat{\boldsymbol{p}}_j^s} (\min_{\boldsymbol{p}_{i,k}^s \in \boldsymbol{p}_i^s} \|\hat{\boldsymbol{p}}_{j,l}^s - \boldsymbol{p}_{i,k}^s\|), \tag{5}$$

where $\boldsymbol{p}_{i,j}^s$ and $\hat{\boldsymbol{p}}_{i,j}^s$ are the $j$-th support and query object-level prototypes in $i$-th frame, respectively. Then, for the frame-level distance matrix $\hat{\boldsymbol{D}} \in \mathbb{R}^{T \times T}$, we find the smallest distance across the frame sequences, which gives a more confident probability of spatial feature matching. Finally, the spatial metric can be formulated as:

$$\mathcal{D}_s = \frac{1}{T} \sum_{i=1}^T (\min_{d_{i,j} \in \hat{\boldsymbol{D}}} \|d_{ij}\|) + \frac{1}{T} \sum_{j=1}^T (\min_{d_{i,j} \in \hat{\boldsymbol{D}}} \|d_{ij}\|). \tag{6}$$

**Few-shot Temporal Metric.** After obtaining the frame-level prototypes of support and query videos in a few-shot task, like in previous works [11, 24], we obtain support-query matching results by applying the temporal alignment metric:

$$\mathcal{D}_t = \texttt{Metric}(\boldsymbol{P}_t, \hat{\boldsymbol{P}}_t), \tag{7}$$

Table 1: **Quantitative comparison results on HMDB51 [31] and UCF101 [32]** (see §4.3). The experiment settings are conducted under the 5-way $K$-shot. "INet-RN50" denotes ResNet-50 pre-trained on ImageNet.

| Method | | Pre-training | HMDB51 | | | UCF101 | | |
|---|---|---|---|---|---|---|---|---|
| | | | 1-shot | 3-shot | 5-shot | 1-shot | 3-shot | 5-shot |
| ARN [10] | [ECCV20] | C3D | 45.5 | - | 60.6 | 66.3 | - | 83.1 |
| OTAM [11] | [CVPR20] | INet-RN50 | 54.5 | 65.7 | - | 79.9 | 87.0 | - |
| TRX [18] | [CVPR21] | INet-RN50 | 53.1 | 66.8 | 75.6 | 78.2 | 92.4 | 96.1 |
| MTFAN [23] | [CVPR22] | INet-RN50 | 59.0 | - | 74.6 | 84.8 | - | 95.1 |
| HyRSM [16] | [CVPR22] | INet-RN50 | 60.3 | 71.7 | 76.0 | 83.9 | 93.0 | 94.7 |
| STRM [15] | [CVPR22] | INet-RN50 | 52.3 | 67.4 | 77.3 | 80.5 | 92.7 | 96.9 |
| CPM [17] | [ECCV22] | INet-RN50 | 60.1 | - | - | 71.4 | - | - |
| HCL [19] | [ECCV22] | INet-RN50 | 59.1 | 71.2 | 76.3 | 82.6 | 91.0 | 94.5 |
| MoLo [70] | [CVPR23] | INet-RN50 | 60.8 | 72.0 | 77.4 | 86.0 | 93.5 | 95.5 |
| GgHM [13] | [ICCV23] | INet-RN50 | 61.2 | - | 76.9 | 85.2 | - | 96.3 |
| CLIP-FSAR [24] | [IJCV24] | CLIP-RN50 | 69.2 | 77.6 | 80.3 | 91.3 | 95.1 | 97.0 |
| CapFSAR [21] | [Arxiv23] | BLIP-ViT-B | 65.2 | - | 78.6 | 93.3 | - | 97.8 |
| CLIP-Freeze [27] | [ICML21] | CLIP-ViT-B | 58.2 | 72.7 | 77.0 | 89.7 | 94.3 | 95.7 |
| CLIP-FSAR [24] | [IJCV24] | CLIP-ViT-B | 75.8 | 84.1 | 87.7 | 96.6 | 98.4 | 99.0 |
| DIST (**Ours**) | | CLIP-ViT-B | $82.6_{\pm 0.3}$ | $87.1_{\pm 0.3}$ | $88.7_{\pm 0.1}$ | $98.3_{\pm 0.2}$ | $99.0_{\pm 0.2}$ | $99.2_{\pm 0.1}$ |

where $\boldsymbol{P}_{\mathrm{t}} \in \mathbb{R}^{T \times C}$ represents query frame-level prototypes, $\hat{\boldsymbol{P}}_{\mathrm{t}} \in \mathbb{R}^{T \times C}$ is support frame-level prototypes, and Metric denotes the OTAM [11] metric by default. We formulate the distance between support and query videos as the weighted sum of the distances obtained by the few-shot spatial metric and few-shot temporal metric:

$$\mathcal{D} = \mathcal{D}_{\mathrm{t}} + \alpha \mathcal{D}_{\mathrm{s}}, \tag{8}$$

where $\alpha$ is a coefficient parameter. Our proposed matching strategy combines the advantages of frame- and object-level prototype matching to cope with appearance- and motion-centric actions.

Following previous works [11, 18, 16], we minimize cross-entropy loss $\mathcal{L}_{\mathrm{CE}}$ over the support-query distances based on the ground-truth labels to end-to-end train DIST. For few-shot inference, total support-query distance in Eq. 8 is employed as logits to produce final query prediction.

# 4 EXPERIMENTS

## 4.1 EXPERIMENTAL SETUP

**Dataset.** We conduct extensive experiments on five datasets, *i.e.*, Kinetics [33], SSv2-full [34], SSv2-small [34], HMDB51 [31], and UCF101 [32]. For SSv2-full [34], SSv2-Small [34] and Kinetics [33], we utilize the split as in CMN [52], with 64, 12, and 24 classes used for train, val, and test, respectively. For HMDB51 [31] and UCF101 [32], we adopt the split setting as in ARN [10], where the 51 classes in HMDB51 are split into 31/10/10 classes for train/val/test, while the 101 classes in UCF101 are split into 70/10/21 classes for train/val/test.

**Evaluation.** Following the official evaluation protocols [13, 11], we use 5-way 1-shot and 5-shot accuracy for evaluation, and report average results over 10,000 tasks randomly selected from test.

## 4.2 IMPLEMENTATION DETAILS

**Network Architecture.** We use CLIP ViT-B [27] as our backbone for a fair comparison with previous methods [24, 21]. By default, the number of spatial attributes $G$ and temporal attributes $L$ are set to 6 and 3, respectively (ablation study in Table 7 of Appendix). The number of object-level prototypes $N$ is 9. The value of parameter $\alpha$ is set to 0.5 (see Fig. 4 (left)).

**Network Training.** Following previous methods [1, 16, 11, 13], we uniformly and sparsely sample $T = 8$ frames of each video to encode video representation. In the training phase, we adopt basic data augmentation, such as random horizontal flipping, cropping, and color jitter. To retain the original pre-trained prior knowledge in the text encoder and reduce the optimization burden, we freeze the text encoder and prevent it from being updated during training. Moreover, we use the Adam [71] optimizer with the multi-step scheduler to train our framework.

**Reproducibility.** DIST is implemented in PyTorch, and all models are trained and tested on two NVIDIA Tesla V100 GPUs with a 32GB memory per card. Full code will be released.

Table 2: **Quantitative comparison results on Kinetics [33] and SSv2-small [34]** (see §4.3). The experiment settings are conducted under the 5-way $K$-shot. "INet-RN50" denotes ResNet-50 pre-trained on ImageNet.

| Method | Pre-training | Kinetics | | SSv2 | | SSv2-small | |
|---|---|---|---|---|---|---|---|
| | | 1-shot | 5-shot | 1-shot | 5-shot | 1-shot | 5-shot |
| ARN [10] [ECCV20] | C3D | 63.7 | 82.4 | - | - | - | - |
| OTAM [11] [CVPR20] | INet-RN50 | 73.0 | 85.8 | 42.8 | 52.3 | 36.4 | 48.0 |
| TRX [18] [CVPR21] | INet-RN50 | 63.6 | 85.9 | 42.0 | 64.6 | 36.0 | 56.7 |
| MTFAN [23] [CVPR22] | INet-RN50 | 74.6 | 87.4 | 45.7 | 60.4 | - | - |
| HyRSM [16] [CVPR22] | INet-RN50 | 73.7 | 86.1 | 54.3 | 69.0 | 40.6 | 56.1 |
| STRM [15] [CVPR22] | INet-RN50 | 62.9 | 86.7 | 43.1 | 68.1 | 37.1 | 55.3 |
| CPM [17] [ECCV22] | INet-RN50 | 73.3 | - | 49.3 | 66.7 | - | - |
| HCL [19] [ECCV22] | INet-RN50 | 73.7 | 85.8 | 47.3 | 64.9 | 38.9 | 55.4 |
| MoLo [70] [CVPR23] | INet-RN50 | 74.0 | 85.6 | 56.6 | 70.6 | 42.7 | 56.4 |
| CLIP-FSAR [24] [IJCV24] | CLIP-RN50 | 87.6 | 91.9 | 58.1 | 62.8 | 52.0 | 55.8 |
| CapFSAR [21] [Arxiv23] | BLIP-ViT-B | 84.9 | 93.1 | 51.9 | 68.2 | 45.9 | 59.9 |
| CLIP-Freeze [27] [ICML21] | CLIP-ViT-B | 78.9 | 91.9 | 30.0 | 42.4 | 29.5 | 42.5 |
| CLIP-FSAR [24] [IJCV24] | CLIP-ViT-B | 89.7 | 95.0 | 61.9 | 72.1 | 54.5 | 61.8 |
| DIST (**Ours**) | CLIP-ViT-B | $\mathbf{92.7}_{\pm0.3}$ | $\mathbf{95.5}_{\pm0.1}$ | $\mathbf{64.2}_{\pm0.2}$ | $\mathbf{75.2}_{\pm0.2}$ | $\mathbf{57.5}_{\pm0.3}$ | $\mathbf{62.5}_{\pm0.1}$ |

Table 3: **Comparison results [34, 32, 31, 33]** (see §4.3) by combining few-shot and zero-shot results.

| Method | HMDB51 | | UCF101 | | Kinetics | | SSv2 | | SSv2-small | |
|---|---|---|---|---|---|---|---|---|---|---|
| | 1-shot | 5-shot | 1-shot | 5-shot | 1-shot | 5-shot | 1-shot | 5-shot | 1-shot | 5-shot |
| CLIP-FSAR [24] [IJCV24] | 77.1 | 87.7 | 97.0 | 99.1 | 94.8 | 95.4 | 62.1 | 72.1 | 54.6 | 61.8 |
| DIST (**Ours**) | **82.6** | **88.7** | **98.3** | **99.2** | **95.6** | **96.0** | **64.6** | **75.8** | **57.5** | **62.5** |

## 4.3 COMPARISON WITH STATE-OF-THE-ARTS

We compare the performance of our DIST with current state-of-the-art FSAR methods on five standard datasets [31, 32, 34, 33] in Table 1 and Table 2. It demonstrates that DIST outperforms all FSAR methods. Specifically, compared to CLIP-FSAR [24] that only uses naive class names as semantic information, our approach achieves better results in multiple datasets and task settings. It indicates our DIST further boosts performance by grasping spatiotemporal-decoupled prior knowledge from LLM to compensate for visual features. Further, the performance margin between DIST and CLIP-FSAR is more significant under low shots. Notably, on HMDB51 [31] and UCF101 [32] datasets, the performance of our DIST on the 5-way 3-shot setting is comparable to the performance of CLIP-FSAR on the 5-way 5-shot setting. We attribute this to the fact that decoupled prompts provide enough background knowledge to compensate for visual features than naive class names especially when visual information is insufficient (*i.e.*, one shot). Furthermore, Table 3 compares our DIST against CLIP-FSAR under another setting, which makes few-shot predictions with the help of zero-shot results. The results show DIST consistently outperforms CLIP-FSAR on each dataset.

## 4.4 ABLATION STUDY

We conduct ablation experiments to evaluate the efficacy of our idea and core model designs. Unless otherwise specified, we adopt CLIP-ViT-B model as default experimental setting.

**Key Component Analysis.** Table 4 summarizes the impact of each module in DIST. Specifically, compared to the baseline, **Temporal Knowledge Compensator (TKC)** (*cf.*§3.5) brings **5.2%**, **2.3%** and **1.9%** performance gains on HMDB51 [31], SSv2-small [33] and UCF101 [32], respectively. This consistent promotion indicates that TKC can enhance the tem-

Table 4: **Impacts of core components** on HMDB51 [31], SSv2-small [33], and UCF101 [32] in the 5-way 1-shot tasks (see §4.4).

| Method Component | HMDB51 | SSv2-small | UCF101 |
|---|---|---|---|
| BASELINE | 75.8 | 53.8 | 96.0 |
| SKC *only* | 77.6 | 55.7 | 96.6 |
| TKC *only* | 81.0 | 56.1 | 97.9 |
| DIST (**Ours**) | **82.6** | **57.5** | **98.3** |

poral awareness of DIST to facilitate accurate matching. In addition, the proposed **Spatial Knowledge Compensator (SKC)** (*cf.*§3.4) improves on the three datasets [31, 32] by **1.8%**, **1.9%** and **0.6%**, respectively, which indicates leveraging spatial prior knowledge can focus on action-related spatial details to boost few-shot performance. Moreover, combining the two modules can further improve performance, indicating the complementarity between spatial and temporal prior knowledge.

**Attribute Injection Manners.** We respectively propose SKC (*cf.*§3.4) and TKC (*cf.*§3.5) to inject spatial/temporal attribute features into visual features. In Table 5a, we study the effect of different temporal/spatial attribute injection manners. "Concat" means that the visual features and attribute features are directly concatenated and then fed into transformers for multimodal fusion like CLIP-

Table 5: **A set of ablation studies on HMDB51** [31] (see §4.4). The adopted network designs are marked in red.

| Spatial Attribute | | Temporal Attribute | | HMDB51 | |
|---|---|---|---|---|---|
| Concat | SKC | Concat | TKC | 1-shot | 5-shot |
| ✓ | | ✓ | | 80.7 | 88.3 |
| | ✓ | ✓ | | 81.0 | 88.6 |
| ✓ | | | ✓ | 81.6 | 88.5 |
| | ✓ | | ✓ | **82.6** | **88.7** |

(a) attribute injection manner

| Spatial Attribute | | Temporal Attribute | | HMDB51 | |
|---|---|---|---|---|---|
| Label | Knowledge | Label | Knowledge | 1-shot | 5-shot |
| ✓ | | ✓ | | 80.0 | 87.3 |
| | ✓ | ✓ | | 81.2 | 88.0 |
| ✓ | | | ✓ | 81.6 | 88.6 |
| | ✓ | | ✓ | **82.6** | **88.7** |

(b) attribute content

| Method | HMDB51 | SSv2-small |
|---|---|---|
| CLIP-FSAR [24] | 75.8 | 53.8 |
| CLIP-FSAR† | 81.0 | 56.1 |
| DIST (**Ours**) | **82.6** | **57.5** |

(c) knowledge compensator

| Spatial Metric | HMDB51 | |
|---|---|---|
| | 1-shot | 5-shot |
| One-to-one matching | 82.4 | 87.8 |
| Bi-MHM [16] | 82.4 | 87.9 |
| **Ours** | **82.6** | **88.7** |

(d) spatial matching metric

| Temporal Metric | HMDB51 | |
|---|---|---|
| | 1-shot | 5-shot |
| CLIP-FSAR (Bi-MHM) [24] | 76.0 | 87.8 |
| **Ours (Bi-MHM)** | **82.7** | **88.9** |
| CLIP-FSAR (OTAM) [24] | 75.8 | 87.7 |
| **Ours (OTAM)** | **82.6** | **88.7** |

(e) temporal matching metric

Figure 4: **Left**: **The impact of the varying fusion parameter** $\alpha$ on HMDB51 [31] in the 5-way 1-shot setting (see §4.4). **Right**: **5-way 1-shot class improvement of DIST** compared to CLIP-FSAR [24] on all class action classes on HMDB51 [31] (see §4.5). Our DIST achieves improvement on all action classes.

FSAR [24]. The experimental results show that our proposed SKC and TKC yields better results, suggesting the effectiveness of our module design.

**Attribute Content.** We investigate the impact of different temporal and spatial attribute content on the performance of our proposed DIST. As shown in Table 5b, we observe that utilizing spatial and temporal prior knowledge generated by LLM consistently performs better than using class names, with $1.2\%$ and $1.6\%$ performance gains in the 1-shot setting on HMDB [31], respectively. In addition, combining spatial and temporal prior knowledge yields better results, which demonstrate different prior knowledge is complementary to others.

**Impact of Knowledge Compensators.** We replace the category labels of CLIP-FSAR [24] with LLM-generated prompts (*i.e.*, CLIP-FSAR†) and report the comparison results in Table 5c. DIST gains larger improvements Compared to CLIP-FSAR†. This suggests our performance gains are not solely due to the usage of LLM-generated prompts, but also due to proposed knowledge compensators which make full use of LLM prompts to compensate for visual features.

**Matching Metrics.** We analyze the impact of different spatial matching metrics in Table 5d. We adopt different spatial matching metrics (*cf*. Eq. 6), including one-to-one matching, Bi-MHM [16], and our proposed spatial metric. One-to-one matching means computing the spatial matching scores of aligned object-level prototypes between the support video and query video. The results show that our proposed spatial metric achieves the best results, suggesting the effectiveness of our proposed spatial matching metric. We also conduct experiments using different temporal matching metrics (*cf*. Eq. 7) on HMDB51 [31]. As shown in Table 5e, our method can adapt to any temporal alignment metric and achieves better performance compared to CLIP-FSAR [24].

**Varying Fusion Parameter** $\alpha$**.** Fig. 4 (left) shows the impact of the varying fusion parameter $\alpha$ (*cf*. Eq. 8) of spatial and temporal matching in the 5-way 1-shot task on HMDB51 [31]. From the results, the optimal value of parameter $\alpha$ is 0.5 for HMDB51.

## 4.5 QUALITY ANALYSIS

**Class-wise Performance Gains.** Fig. 4 (right) shows 5-way 1-shot class-wise performance gains obtained by our DIST over CLIP-FSAR [24] on HMDB51 [31]. Notably, our DIST achieves performance gains in all action classes. We also observe that DIST achieves gains above $10\%$ for classes

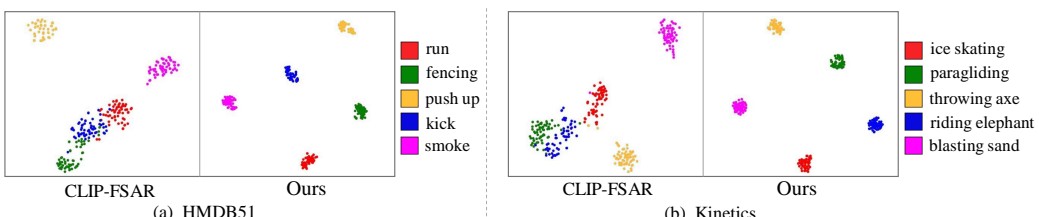

Figure 5: **T-SNE visualization** [72] under the 5-way setting on HMDB51 and Kinetics (see §4.5).

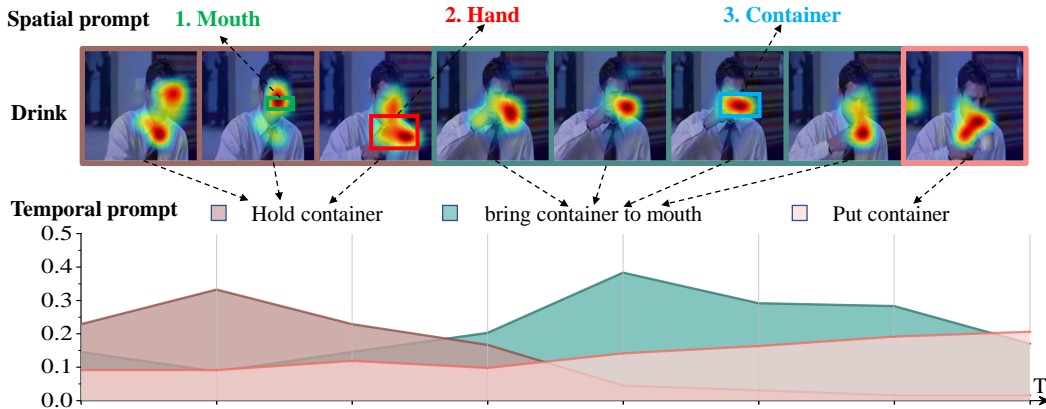

Figure 6: **Visualization of spatial and temporal prompts** under the 5-way 1-shot setting (see §4.5). The spatial prompts are shown as highlighted response areas in each frame. We also show cross-attention temporal prompt weights of Eq. 4 in a line graph.

such as *run*, *pour*, *kick ball*, *etc.*. It indicates that the spatiotemporal-decoupled prior knowledge can easily include objects involved in these actions and capture action-related dynamic information.

**Visualization of Feature Distribution.** To further qualitatively analyze the changes in feature distribution after incorporating spatiotemporal-decoupled prior knowledge, we follow previous methods [24, 15] to visualize the feature distribution of CLIP-FSAR [24] (only category name as semantic information) and our algorithm DIST in Fig. 5. We observe that after utilizing action-related prior knowledge, our method shows more compact intra-class feature distributions and more discriminative inter-class features. As shown in Fig. 5 (a), the three classes "run", "fencing", and "kick" become clearly distinguishable from each other after injecting decoupled attributes into visual features.

**Visualization of Spatial and Temporal attributes.** To analyze the role of spatial and temporal attributes in DIST, Fig. 6 displays the visualization results of these attributes. As seen, the attention maps of our DIST focus more on action-related objects and reduce attention to the background and unrelated objects. This demonstrates our DIST grasps prior knowledge provided by spatial attributes to capture spatial details. Then, we calculate the cross-attention scores between temporal attributes and frames according to Eq. 4. It can be seen that different temporal attributes have different weights on the frame sequences, which proves that our DIST can learn temporal relations and capture dynamic semantics. For example, the temporal attribute "Hold container" has larger weights on the first three frames, which indicates these frames may correspond to dynamic semantics implied by the temporal attribute. See more examples in §C.2 of Appdendix.

# 5 CONCLUSION

In this work, we propose a novel yet effective DIST framework for FSAR, which is the first work to grasp spatiotemporal-decoupled prior knowledge from LLM to compensate for visual features. In particular, we design Spatial/Temporal Knowledge Compensators to learn object- and frame-level prototypes, so as to capture fine-grained spatial details and dynamic semantics. Experimental results demonstrate that our DIST achieves state-of-the-art performance on four standard benchmarks. However, this is the first cursory exploration of leveraging spatiotemporal-decoupled prior knowledge in FSAR. Though the first step is not always elegant, exploring additional attempts to leverage richer prior knowledge provided by LLMs promises intriguing prospects for the future.

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

- §A provides additional implementation details.
- §B provides the pseudo-code of spatial and temporal feature matching.
- §C introduces more quantitative and qualitative experiment results.
- §D shows more additional examples of spatio-temporal knowledge.
- §E discusses our limitations and social impact.

## A  ADDITIONAL IMPLEMENTATION DETAILS

### A.1  DATASET DETAILS

We compare the proposed DIST with top-leading methods on four few-shot benchmarks, including Kinetics [33], HMDB51 [31], UCF101 [32], and SSv2-small [34].

- **HMDB51** [31] contains 51 action classes and has 6,766 video clips. Following the setting of previous methods [52, 24] for few-shot action recognition, we divide them into 31, 10, and 10 action classes as train/val/test. The constructed dataset has 31 action classes with 4,280 videos, 10 action classes with 1,194 videos, and 10 action classes with 1,292 videos for meta-training, meta-validation, and meta-testing.

- **Kinetics** [33] contains 400 action classes and has 306,245 video clips. Following the setting of previous methods [52, 16] for few-shot action recognition, we select 100 classes and divide them into 64, 12, and 24 action classes as train/val/test. The constructed dataset has 64 action classes with 6,389 videos, 12 action classes with 1,199 videos, and 24 action classes with 2,395 videos for meta-training, meta-validation, and meta-testing.

- **UCF101** [32] contains 101 action classes and has 13,320 video clips. Following the setting of previous methods [52, 24] for few-shot action recognition, we divide them into 70, 10, and 21 action classes as train/val/test. The constructed dataset has 70 action classes with 9,154 videos, 10 action classes with 1,421 videos, and 10 action classes with 2,745 videos for meta-training, meta-validation, and meta-testing.

- **SSv2-small** [34] contains 174 action classes and has 220,847 video clips. Following the setting of previous methods [11, 24] for few-shot action recognition, we select 100 classes and divide them into 64, 12, and 24 action classes as the meta-training, meta-validation, and meta-testing set, respectively.

### A.2  IMPLEMENTATION DETAILS

Following previous methods [13, 24, 19, 16], we uniformly sample $T = 8$ frames from input videos, which are scaled to a height of 256. In the training phase, we adopt basic data augmentation, such as random horizontal flipping, cropping, and color jitter. In contrast, only a center crop is used during the testing phase. We use the PyTorch library to train our DIST on two Tesla v100 GPUs. Moreover, our framework uses the Adam optimizer [71] with the multi-step scheduler to train our model. The total number of training steps is set to 10. Table 6 shows the same settings of hyperparameters with CLIP-FSAR [24] in a multi-step scheduler for various datasets. In this table, **lr** represents the learning rate, **steps** indicates the number of steps to change the learning rate, **iter_st** refers to the number of iterations per step, and **lr_st** denotes the multiplication factor for updating the learning rate at each changing step.

## B  PSEUDO CODE OF SPATIAL AND TEMPORAL MATCHING

Algorithm 1 provides the pseudo-code of spatial and temporal feature matching in a PyTorch-like style. With acquired action-related prior knowledge, we decompose video matching process into complementary object-level and frame-level prototype matching. To guarantee reproducibility, full code will be released.

Table 6: The settings of hyperparameters in multi-step scheduler in various datasets (see §A.2).

| Dataset | lr | steps | iter_st | lr_st |
|---|---|---|---|---|
| Kinetics [33] | $1e-5$ | [0,6,9] | 1000 | [1,0.1,0.01] |
| HMDB51 [31] | $1e-5$ | [0,4,6] | 2800 | [1,0.1,0.01] |
| UCF101 [32] | $2e-6$ | [0,4,6] | 2400 | [1,0.1,0.01] |
| SSv2-small [34] | $5e-5$ | [0,4,6] | 8000 | [1,0.1,0.01] |

**Algorithm 1** Pseudo-code of object-level and frame-level prototype matching in a PyTorch-like style.

```
# q_obj_p: query object prototype (B_q x T x N x C)
# s_obj_p: support object prototype (B_s x T x N x C)
# q_frm_p: query frame prototype (B_q x T x C)
# s_frm_p: support frame prototype (B_s x T x C)
# B_q: number of query videos
# B_s: number of support videos
# T: number of frames
# N: number of object prototypes in each frame
# OTAM: temporal metric

def matching(q_obj_p, s_obj_p, q_frm_p, s_frm_p):
    # Object-level prototype matching
    q_obj_p = q_obj_p.reshape(B_q x T x N, C) # (B_q x T x N) x C
    s_obj_p = s_obj_p.reshape(B_s x T x N, C) # (B_s x T x N) x C
    obj_sim = cos_sim(q_obj_p, s_obj_p) # (B_q x T x N) x (B_s x T x N)
    obj_dist = 1 - obj_sim # (B_q x T x N) x (B_s x T x N)
    obj_dist = rearrange(obj_dist) # (B_q x T) x (B_s x T) x N x N
    obj_fdist = obj_dist.min(3)[0].sum(2) + obj_dist.min(2)[0].sum(2) #
        (B_q x T) x (B_s x T)
    obj_fdist = rearrange(obj_fdist) # B_q x B_s x T x T
    obj_logits = obj_fdist.min(3)[0].mean(2) + obj_fdist.min(2)[0].mean
        (2) # B_q x B_s

    # Frame-level prototype matching
    q_frm_p = q_frm_p.reshape(B_q x T, C) # (B_q x T) x C
    s_frm_p = s_obj_p.reshape(B_s x T, C) # (B_s x T) x C
    frm_sim = cos_sim(q_frm_p, s_frm_p) (B_q x T) x (B_s x T)
    frm_dist = 1 - frm_sim # (B_q x T) x (B_s x T)
    frm_dist = rearrange(frm_dist) # B_q x B_s x T x T
    frm_logits = OTAM(frm_dist) # B_q x B_s

    return obj_logits, frm_logits
```

# C  MORE EXPERIMENT RESULTS

## C.1  IMPACT OF DIFFERENT NUMBERS OF PROMPTS

We conduct experiments on various prompt configurations in Table 7. First, we conduct experiments with 3, 6, and 12 spatial attributes. We can observe that the performance reaches its peak at $G = 6$, possibly because too many spatial attributes may introduce local noise, while too few spatial attributes cannot afford enough spatial information. Furthermore, the best result is obtained when the number of temporal attributes is $L = 3$. We speculate that too few temporal attributes may fail to

Table 7: The ablation study on HMDB51 [31] to investigate the configuration of the number of spatial attributes $G$ and temporal attributes $L$ (see §C.1).

| {G,L} | HMDB51 | |
|---|---|---|
| | 1-shot | 5-shot |
| {3,3} | 82.4 | 88.5 |
| {12,3} | 82.3 | 88.3 |
| {6,1} | 81.4 | 88.0 |
| {6,6} | 81.5 | 88.0 |
| {6,3} | **82.6** | **88.7** |

adequately convey the temporal changes in actions while too many temporal attributes often contain noisy temporal information, which leads to performance degradation. Therefore, we set $G = 6$ and $L = 3$ by default.

## C.2 More Visualization Examples of Spatio-temporal Attributes

We show more visualization results of spatial and temporal prompts in our DIST in Fig. 7. The attention maps of our DIST focus more on action-related objects and reduce attention to the background and unrelated objects. This demonstrates our DIST grasps prior knowledge provided by spatial prompts to capture spatial details. Then, we calculate the cross-attention scores between temporal prompts and frames. It can be seen that different temporal prompts have different weights on the frame sequences, proving that our DIST can learn temporal relations and understand dynamic semantics. This further illustrates that our DIST can capture action-related spatial details and dynamic temporal information.

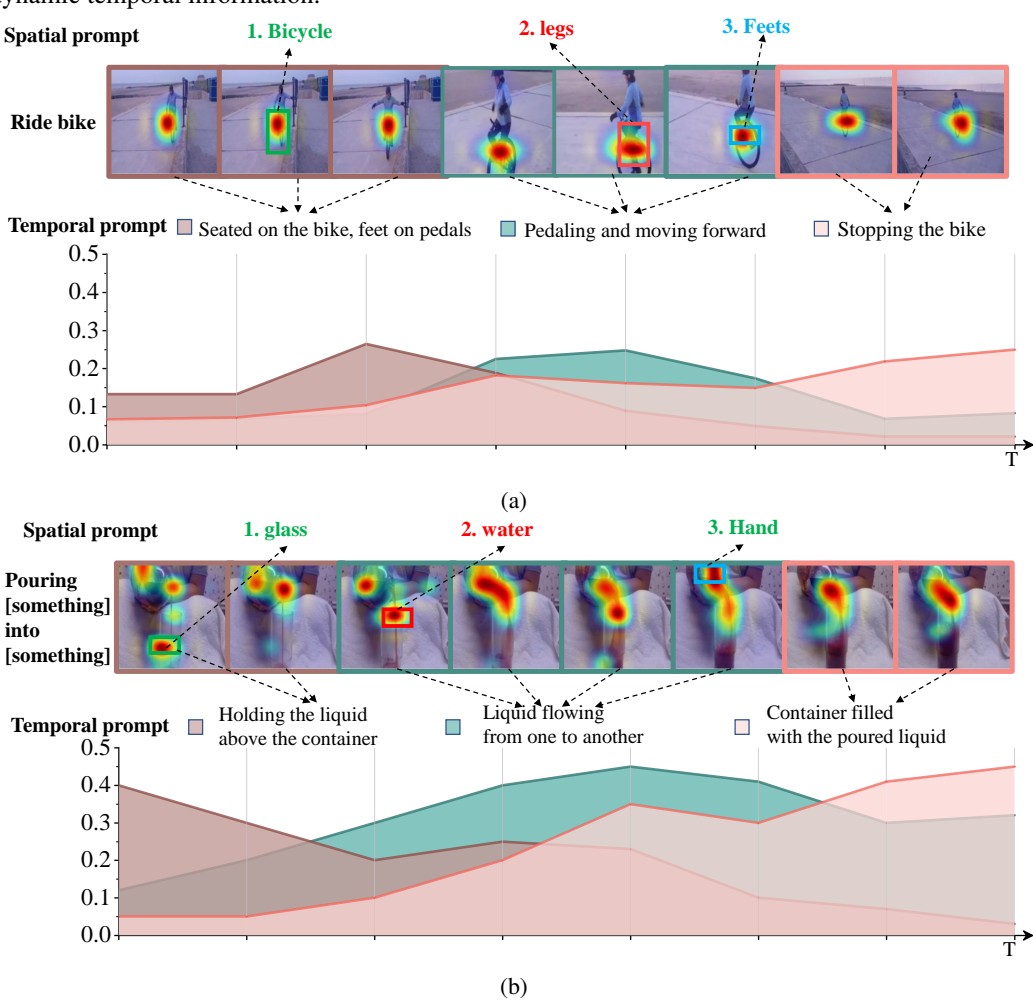

Figure 7: More visualization results of spatial and temporal prompts in our DIST under the 5-way 1-shot setting. The spatial prompts are shown as highlighted response areas in each frame. Meanwhile, we show cross-attention temporal prompt weights in a line graph. See §C.2 for more details.

## C.3 Model Efficiency Analysis

To analyze the effectiveness of training and inference, we list comparison results with SOTA CLIP-FSAR [24] in terms of parameters, FLOPs, GPU memory, and inference speed in Table 8. We choose ViT-B as our visual encoder. As shown in Table 8, compared to SOTA CLIP-FSAR, our additional

Table 8: Complexity analysis for 5-way 1-shot HMDB51 [31] and Kinetics [33] evaluation. Here, we report Params, FLOPs, GPU memory, and Speed for each model. "Acc$^1$" and "Acc$^2$" are the accuarcy on HMDB51 and Kinetics, respectively. See §C.3 for more details.

| Method | Params | FLOPs | Memory | Speed | Acc$^1$ | Acc$^2$ |
|---|---|---|---|---|---|---|
| CLIP-FSAR | 89.3M | 901.9G | 13.8G | 36.7ms | 75.8 | 89.7 |
| DIST(ours) | 97.2M | 902.1G | 14.1G | 40.9ms | 82.6 | 92.7 |

Table 9: Some examples of action categories and their corresponding spatio-temporal decoupled knowledge generated by LLMs.

| Category | Spatial knowledge | | Temporal knowledge |
|---|---|---|---|
| dive | 1. swimming pool 
 3. arms 
 5. pool | 2. diving board 
 4. legs 
 6. cliff | 1. Stand at the diving board. 
 2. Jump into the water. 
 3. Resurface in the water. |
| kick ball | 1. football 
 3. feet 
 5. playground | 2. playground ball 
 4. legs 
 6. park | 1. Stand near a ball. 
 2. Kick the ball. 
 3. The ball in motion, kick ended. |
| archery | 1. bow 
 3. hands 
 5. archery Range | 2. arrows 
 4. arms 
 6. outdoor Field | 1. Bow, arrow, stance set. 
 2. Drawing the bow, aiming. 
 3. The arrow hits the target. |
| brush teeth | 1. toothbrush 
 3. hands 
 5. bathroom | 2. toothpaste 
 4. mouth 
 6. bedroom | 1. A person stands in front of the sink. 
 2. Brushing teeth using a toothbrush and toothpaste. 
 3. Rinsing mouth and cleaning toothbrush. |
| apply eye makeup | 1. eyeshadow palette 
 3. hands 
 5. face | 2. makeup brush 
 4. eye 
 6. dressing Room | 1. Clean face, makeup tools ready. 
 2. Applying eye shadow, liner, and mascara. 
 3. Finished eye makeup, enhanced eyes. |
| air drumming | 1. air drumming app 
 3. hands 
 5. bedroom | 2. drumsticks 
 4. arms 
 6. music studio | 1. Tapping hands in rhythm. 
 2. Mimicking drumming motions. 
 3. Slowing down, stopping drumming movements. |
| blowing out candles | 1. candle 
 3. hands 
 5. birthday party | 2. cake 
 4. mouth 
 6. cake shop | 1. Lit candles on a surface. 
 2. Blowing air to extinguish the flames. 
 3. Candles are extinguished, and the celebration ends. |
| bowling | 1. bowling ball 
 3. hand 
 5. bowling alley | 2. bowling shoes 
 4. arm 
 6. entertainment center | 1. Holding the bowling ball, ready to throw. 
 2. Releasing the ball, rolling down the lane. 
 3. Pins knocked down or remaining. |

overhead is minimal. However, our DIST can bring 6.8% and 3.0% accuracy improvements on HMDB51 and Kinetics over CLIP-FSAR, respectively.

## D  ADDITIONAL EXAMPLES OF SPATIO-TEMPORAL KNOWLEDGE

In Table 9, we display some additional examples of action categories and their corresponding spatio-temporal knowledge. This representative knowledge is automatically generated by LLMs.

## E  DISCUSSION

### E.1  LIMITATION ANALYSIS

DIST relies on large language models to generate high-quality spatial and temporal prompts, which affect the final performance. In addition, relying solely on a fixed number of spatial and temporal prompts may not be optimal for every category. An adaptive approach, customizing the number of prompts for each category, would likely be more effective. In the future, we will also explore a more unified and effective way to inject spatial and temporal prompts into visual features.

### E.2  SOCIAL IMPACT

This work proposes a novel framework to leverage spatiotemporal-decoupled prior knowledge from LLM to compensate for visual features, so as to achieve more accurate few-shot matching. Our framework has demonstrated its effectiveness over multiple common benchmarks. On the positive side, the approach advances few-shot action recognition accuracy, and is valuable for real-world applications in the automated understanding of human actions from videos [73, 74], *e.g.*, human-robotic interaction in elderly care facilities [74], behavior analysis for nursing procedures [73]. For potential negative social impact, our DIST struggles to understand human actions across varying domains, which is a common limitation shared by all few-shot action recognition methods [13, 16, 15]. Hence in the future, we will broaden the few-shot action recognition capability to generalize across varying domains.

