# OpenReview forum: "Spatio-temporal Decoupled Knowledge Compensator for Few-Shot Action Recognition"
_ICLR.cc/2025/Conference — ICLR 2025 Conference Withdrawn Submission_

### Official Review · Reviewer_9xmL · 2024-10-16

**Soundness:** 1
**Presentation:** 3
**Contribution:** 2
**Rating:** 5
**Confidence:** 4

**Summary:**

This paper proposes an innovative decoupling framework named DIST for few-shot action recognition (FSAR). The framework utilizes the decoupled spatiotemporal knowledge provided by large language models (LLMs) to learn expressive prototypes at multiple granularities, thereby capturing fine-grained spatial details and dynamic temporal information.

**Strengths:**

1. Comprehensive experimental setup.
2. Outstanding experimental results, achieving significant improvements on various datasets.
3. Clear and aesthetically pleasing images and tables.

**Weaknesses:**

**1. The method is not sufficiently reasonable.**

The authors propose to utilize information provided by large models after decoupling space and time, but the method of providing this information is too naive.

1.1 The authors do not consider whether irrelevant objects will introduce noise when providing spatial information. The method would be more reasonable if an object detection model could be used to check whether the supplementary object information exists. The authors should conduct a noise analysis experiment.

1.2 The authors' method of providing supplementary temporal information is also not reasonable enough. In SSV2, there are many categories such as "moving an object from the left to the right" and "moving an object from the right to the left". Under the authors' proposed method, the temporal information extracted from these two categories would be very similar, but in fact, there is a significant difference in the temporal information between the two. A further analysis should be conducted.

**2. The statement of the method's contribution needs to be modified.**

The authors write in the paper Sec.3.6, "We propose an object-level prototype matching strategy based on the bidirectional Hausdorff Distance," but in reality, this method was proposed in the 2022 CVPR HyRSM paper[1]. I believe the "we propose" description should be modified, and the citation should be added. Otherwise, the authors may explain the difference further in the paper [1].

[1] Wang X, Zhang S, Qing Z, et al. Hybrid relation guided set matching for few-shot action recognition[C]//Proceedings of the IEEE/CVF conference on computer vision and pattern recognition. 2022: 19948-19957.

**Questions:**

The performance of this work is outstanding, but the rationality of the method is not sufficient.

---

### Official Review · Reviewer_BpBh · 2024-10-28

**Soundness:** 3
**Presentation:** 3
**Contribution:** 3
**Rating:** 5
**Confidence:** 4

**Summary:**

This paper proposes the DIST framework for few-shot action recognition, leveraging spatial and temporal knowledge from large language models (LLMs) to enhance class prototypes. The Spatial and Temporal Knowledge Compensators capture object- and frame-level action features by combining LLM knowledge with visual cues. Extensive experiments demonstrate the effectiveness of DIST.

**Strengths:**

1. The approach of leveraging spatial and temporal attributes to enhance few-shot action recognition is well-motivated.
2. Extensive experiments validate the effectiveness of DIST and demonstrate the contribution of each component.

**Weaknesses:**

1. Some experimental settings and analyses are unclear and could be confusing. Refer to the "Questions" section for details.
2. Some previous works have also utilized LLMs to generate attributes or concepts for action recognition or action representation learning [1,2]. The authors should discuss the key differences in how DIST leverages the generated attributes compared to these works and explain why DIST is better in FSAR.

[1] OST: Refining Text Knowledge with Optimal Spatio-Temporal Descriptor for General Video Recognition. CVPR2024.

[2] Language-based Action Concept Spaces Improve Video Self-Supervised Learning. Neurips2023.

Minor points:
1. The abbreviation "PKC" in Line 022 might be incorrect.

**Questions:**

1. In Table 4, what are the detailed settings for BASELINE? Without the Spatial Knowledge Compensator (SKC) and Temporal Knowledge Compensator (TKC), how are the prototypes obtained?
2. To validate the spatial attributes and temporal attributes help to learn spatial and temporal features, has the authors conducted experiments using only spatial attributes or temporal attributes in both compensators? Or using spatial attributes in TKC and temporal attributes in SKC? The purpose of such experiments is to show the pure improvements of using spatial/temporal attributes.
3. In Table 5(b), the HMDB51 1-shot performance using only labels is 80.0, significantly higher than CLIP-FSAR, which also uses only labels. What accounts for this large performance gap?
4. In Table 5(a), DIST scores 80.0 with only labels and improves to 82.6 when using attributes. In Table 5(c), CLIP-FSAR scores 75.8 with labels and improves to 81.0 with LLM-generated prompts. The improvement for CLIP-FSAR when using LLM generated attributes is larger than that for DIST. Why is this the case? Does this suggest that CLIP-FSAR leverages attributes more effectively?
5. Is the CLIP visual encoder initialized with pre-trained CLIP parameters? How does fixing the visual encoder impact performance?

---

### Official Review · Reviewer_QsrR · 2024-10-31

**Soundness:** 3
**Presentation:** 3
**Contribution:** 2
**Rating:** 5
**Confidence:** 5

**Summary:**

This work proposes a new method to address few-shot action recognition based on Language-Image Pretrained models (the work is based on CLIP). The main idea of the work is introducing decomposed text information for supervision, which is different from previous works those mainly adopt action class name. The work proposes to generate different types of text attributes by GPT, i.e., socalled spatial and temporal text attributes in the paper. By leveraging the generated text attributes, the proposed model can enhance spatial and temporal features of an action video. Extensive experiments demonstrate the effectiveness of the proposed model.

**Strengths:**

1. The idea that leveraging different types of text descriptions generated by LLMs is somewhat novel.
2. The performance of the proposed model is good, which is much better than previous models.
3. The authors also conduct detailed analysis to the proposed model.

**Weaknesses:**

1. The proposed model conduct a decomposition according to the GPT outputs, i.e., generated texts about action-related objects and action states. I wonder if there are some overlapping between between object texts and state texts for an action category. More importantly, how can the authors guarantee that action representations learned by the supervision of socalled action states encode temporal information.
2. Please provide some quantitative analysis to demonstrate that the model can exactly extract spatial and temporal information according to the supervision of object texts and state texts.
3. In my understanding, I think the main innovations of this work is not specific to the few-shot setting. Could you provide some experiments in other settings, e.g., base-to-novel zero-shot transfer, open-vocabulary action classification.
4. How the number of attributes affect the model performance? In Table 7, the authors provide a few experiments to demonstrate this. But I want to see more results. Also, I wonder if the authors consider the noise issue from the LLM-generated attribute texts.
5. The writing of the manuscript is not very clear in the method part. For example, how to extract the feature $\hat{P_s}$ in Section 3.6.

**Questions:**

1. The authors do not consider spatial attributes in addition to objects, such as body parts. But it is OK to me.

---

### Official Review · Reviewer_UKP9 · 2024-11-04

**Soundness:** 3
**Presentation:** 3
**Contribution:** 3
**Rating:** 6
**Confidence:** 2

**Summary:**

The authors design a novel framework named DIST for the FSAR task, which mainly contains two compensators to process temporal and spatial features individually. Additionally, the authors use LLM to generate fine-grained descriptions that extend the semantic information.

**Strengths:**

It's a creative attempt to calculate temporal/spatial prototypes with semantic features, emphasizing different aspects. This work achieves good performance on these datasets.

**Weaknesses:**

In the representation of temporal knowledge, there are some noun phrases that incorporate words derived from spatial knowledge. This makes the modeling of the temporal domain less pure, and perhaps it would be better to opt for simpler verbs only.

**Questions:**

Especially on the ssv2 dataset, the ambiguity of the labels makes it extremely risky for GPT to produce descriptions and introduce irrelevant content. These can hardly avoid having a negative impact on the outcome, so whether controlling the number of generated words alone can minimize the impact?

---

### Note · Authors · 2024-11-14

I have read and agree with the venue's withdrawal policy on behalf of myself and my co-authors.